# Chlorobenzene Oxidation over Phosphotungstic-Acid-Coated Cerium Oxide: Synergistic Effect of Phosphotungstic and Cerium Oxide and Inhibition Mechanism of Sulfur Dioxide

Keyu Jiang, Leyuan Dong, Qi Shen, Wei Wu, Xue Wu, Jian Mei * and Shijian Yang

School of Environment & Ecology, Jiangnan University, Wuxi 214122, China
* Correspondence: jsjhmj@126.com

**Abstract:** Ce–based catalysts exhibit a poor stability and activity in chlorinated volatile organic compound (Cl–VOC) oxidation due to their rapid Cl poisoning. Herein, phosphotungstic acid (HPW) was coated on $CeO_2$ to improve its activity and stability for chlorobenzene (CB) oxidation. The HPW coating not only promoted CB adsorption onto $CeO_2$, but also provided Brønsted acid sites to $CeO_2$ for Cl species removal as HCl, thus avoiding Cl poisoning. Hence, a synergistic effect of $CeO_2$ and HPW on HPW/$CeO_2$ was observed, resulting in superior CB oxidation activity and stability. Additionally, to improve the sulfur resistance of the catalyst, the inhibition mechanism of $SO_2$ on CB oxidation by HPW/$CeO_2$ was explored. HPW/$CeO_2$ was prone to sulfation due to the formation of $Ce_2(SO_4)_3$ from the reaction of $SO_2$ and $CeO_2$. Thus, the oxidation ability of HPW/$CeO_2$; the amount of adsorption sites for CB adsorption; and the amounts of $Ce^{4+}$ bonded with $O^{2-}$, lattice oxygen species, and adsorbed oxygen species were decreased by $SO_2$. Meanwhile, $SO_2$ competed with CB for the adsorption sites on HPW/$CeO_2$. Therefore, CB oxidation by HPW/$CeO_2$ was remarkably restrained by $SO_2$. The present work promotes further work on Cl–VOC removal by Ce-based catalysts for anti-$SO_2$ poisoning modification in the future.

**Keywords:** chlorobenzene oxidation; phosphotungstic-acid-coated cerium oxide; synergistic effect; sulfur dioxide inhibition

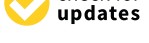



## 1. Introduction

Chlorinated volatile organic compounds (Cl–VOCs) are a type of volatile organic compounds that pose a great threat to human health and the environment due to their persistent toxicity and resistance to biodegradation [1,2]. To combat the detrimental effects of Cl–VOCs, catalytic oxidation has emerged as a promising strategy [3]. This approach involves the use of a catalyst to initiate chemical reactions that convert Cl–VOCs into less harmful substances. However, the success of catalytic oxidation relies heavily on the performance of the catalyst. A well–designed catalyst for Cl–VOC oxidation should possess an exceptional overall performance, which includes an excellent activity, stability, selectivity, and versatility. However, in industrial applications, the selectivity and stability of the catalyst for Cl–VOC oxidation are considered more significant than its activity [4,5]. This is because the challenges posed by Cl poisoning, which results in catalyst deactivation, and the generation of more dangerous polychlorinated compounds cannot be ignored. Therefore, the development of catalysts that can maintain a high stability and selectivity while effectively eliminating Cl–VOCs should be prioritized in order to mitigate the environmental and health hazards associated with these compounds.

To date, catalysts that have been exploited for the oxidation of Cl–VOCs consist of solid acids, noble metals, and transient metal oxides, depending on their active components [6,7]. The use of solid acids for Cl–VOC oxidation usually requires higher temperatures, resulting in a higher energy consumption and the potential production of toxic dioxins [8]. Noble metals, on the other hand, have shown great activity in Cl–VOC oxidation, but they are still

prone to Cl-induced activation and the formation of polychlorinated by-products with a high selectivity [9]. Transient metal oxides, such as $VO_x$ and $CrO_x$, have also demonstrated remarkable activity for Cl–VOC oxidation due to their high oxidizing ability. However, they tend to become deactivated over time due to the gradual loss of activation caused by the formation of chlorinated metal oxides [10,11]. Therefore, the development of well-designed catalysts that can overcome these aforementioned shortcomings is still a challenge.

$CeO_2$ with a fluorite structure is a material that can release oxygen when it is in reducing conditions, effectively storing oxygen in rich oxygen environments [12]. Because of this unique property, $CeO_2$ is extensively utilized in catalysis [13]. Nevertheless, one major drawback of $CeO_2$ is its poor activity and stability when it comes to the oxidation of Cl–VOCs due to rapid Cl poisoning [14]. To overcome this issue, one approach to improving the catalyst's resistance to Cl poisoning in Cl–VOC oxidation is through the introduction of other metals, which enhances its oxidizing ability. Zhou et al. discovered that the incorporation of Cr into $CeO_2$ significantly increases the catalyst's oxidizing ability [15]. $Ce_4Cr_1$ exhibited excellent activity and stability when it came to 1,2–dichloroethane oxidation. Another method to enhance the catalyst's ability to resist Cl poisoning in Cl–VOC oxidation is through the introduction of acids, which can transform Cl species into HCl. This transformation leads to the enhancement of chlorine desorption from the catalyst, preventing chlorination reactions. Gutiérrez–Ortiz et al. revealed that treating $Ce_{0.5}Zr_{0.5}O_2$ with $H_2SO_4$ significantly increased its acidity [16]. This treatment resulted in the catalyst exhibiting excellent stability and activity in 1,2-dichloroethane oxidation.

Phosphotungstic acid (HPW, $H_3PW_{12}O_{40}$) is a type of acid with a Keggin structure [17]. This unique structure gives it a high acidic strength, making it highly effective in various acid-catalyzed reactions [18]. In industrial flue gas, $SO_2$ is often present along with Cl–VOCs [19]. Unfortunately, the presence of $SO_2$ tends to inhibit the oxidation of Cl–VOCs by Ce-based catalysts [20]. The mechanism behind this inhibition is not well understood, which has hindered the optimization of Ce-based catalysts for Cl–VOC oxidation. In this work, HPW was coated on $CeO_2$ to further enhance its activity and stability in the oxidation of CB (a model compound for Cl–VOCs), and the synergistic effect of HPW and $CeO_2$ on CB oxidation was investigated. Moreover, the inhibition mechanism of $SO_2$ on CB oxidation by HPW/$CeO_2$ was also explored.

## 2. Experimental Section

### 2.1. Catalyst Preparation

$CeO_2$ was obtained by calcinating $Ce(NO_3)_3 \cdot 6H_2O$ at 300 °C for 2 h in air. After soaking the pulverous $CeO_2$ in an HPW (purchased from Sinopharm Group Chemical Reagent Co. Ltd., Shanghai, China) solution for 1 h, it was centrifuged and washed multiple times. Afterwards, the particles were calcined at 500 °C for 3 h to obtain HPW/$CeO_2$. Lastly, HPW/$CeO_2$ was treated with 500 ppm $SO_2$ at 450 °C for 10 h with a weight hourly space velocity (WHSV) of $6.0 \times 10^4$ $cm^3$ $g^{-1}$ $h^{-1}$ to obtain sulfated HPW/$CeO_2$.

### 2.2. Catalytic Performance Evaluation

The catalytic performance of CB oxidation was evaluated in a fixed–bed quartz reactor at 250–450 °C in step mode with a 30 min plateau at a given temperature. The catalyst mass was generally 200 mg and the total flow rate of the gas was 200 mL $min^{-1}$, resulting in a WHSV of 60,000 $cm^3$ $g^{-1}$ $h^{-1}$. The simulated flue gas typically contained 100 ppm CB, 5% $O_2$, 100 ppm $SO_2$ (during use), 5% $H_2O$ (during use), and $N_2$ balance. To measure the concentrations of CB, HCl, CO, and $CO_2$ at the outlet of the reactor, a ThermoFisher IGS infrared gas analyzer (Waltham, MA, USA) was used. The catalytic efficiency was evaluated based on several parameters, including the CB conversion efficiency, HCl selectivity, and $CO_x$ selectivity (which includes both CO and $CO_2$). These parameters were calculated using specific equations:

$$\text{CB conversion} = \frac{[\text{CB}]_{\text{in}} - [\text{CB}]_{\text{out}}}{[\text{CB}]_{\text{in}}} \times 100\% \tag{1}$$

$$\text{HCl selectivity} = \frac{[\text{HCl}]_{\text{out}}}{[\text{CB}]_{\text{in}} - [\text{CB}]_{\text{out}}} \times 100\% \tag{2}$$

$$\text{CO}_x \text{ selectivity} = \frac{[\text{CO}_x]_{\text{out}}}{6([\text{CB}]_{\text{in}} - [\text{CB}]_{\text{out}})} \times 100\% \tag{3}$$

### *2.3. Catalyst Characterization*

X-ray diffraction patterns (XRD) were measured on an X-ray diffractometer (Bruker–AXS D8 ADVANCE, Billerica, MA, USA), and the diffractograms were obtained in the 2θ range of 10°–80°, with a scanning velocity of 10°/min. The Brunauer–Emmett–Teller (BET) surface area was determined via a $N_2$ adsorption analyzer (Quantachrome 2200e, Boynton Beach, FL, USA). Before each experiment, the samples were pretreated under vacuum at 200 °C for 3 h [21]. X-ray photoelectron spectra (XPS) were measured on an X-ray photoelectron spectroscope (ThermoFisher ESCALAB 250) using Mg Ka radiation as the excitation source; the binding energy was calibrated using the C 1s line at 284.8 eV as an internal standard. X-ray fluorescence (XRF) measurements were carried out on an X-ray fluorescence analyzer (XRF, ThermoFisher ARL). Differential thermal analysis (DTA) was performed on a Netzsch STA 409 PC thermal analyzer (Selb, Germany). The temperature-programmed reduction of $H_2$ ($H_2$–TPR) was measured via an Autochem II 2920 chemical adsorption analyzer of Micromeritic (Norcross, GA, USA). Scanning transmission electron microscopy (STEM) and energy dispersive spectrometer (EDS) mapping were performed on a JEOL JEM–F200 (URP) instrument (Tokyo, Japan).

The temperature-programmed desorption of CB (CB–TPD) was conducted on the same fixed-bed quartz reactor that was used for the catalytic performance evaluation. A total of 200 mg of catalysts was firstly with by 200 mL min$^{-1}$ of 5% $O_2$/$N_2$ at 400 °C for 1 h, and then cooled to 50 °C. Afterwards, the catalysts were exposed to 100 ppm CB for 1 h. Lastly, the catalysts were purged by 100 mL min$^{-1}$ of $N_2$ from 50 to 600 °C at a heating rate of 10 °C min$^{-1}$.

In situ diffuse reflectance infrared Fourier transform spectroscopy (DRIFTS) was conducted on a Fourier transform infrared spectrometer (Thermo Fisher, Nicolet iS50) equipped with an MCT detector. The spectra were collected at a resolution of 4 cm$^{-1}$ and over 32 scans.

## 3. Results and Discussion

### *3.1. Performance for CB Oxidation*

$CeO_2$ showed poor activity in CB oxidation, with a $T_{90}$ value (the temperature corresponding to 90% conversion efficiency) much larger than 450 °C (Figure 1a). However, $HPW/CeO_2$ showed excellent activity in CB oxidation, and its value of $T_{90}$ was only approximately 340 °C (Figure 1a), which was significantly lower than that of $CeO_2$. This finding suggests that the activity of $CeO_2$ for CB oxidation was significantly improved after coating with HPW.

The ideal products of CB oxidation are generally HCl, $CO_x$, and $H_2O$; thus, the selectivities of $CeO_2$ and $HPW/CeO_2$ toward HCl and $CO_x$ were investigated. Figure 1b shows that $CeO_2$ exhibited a low selectivity toward HCl at 200–450 °C, and its HCl selectivity was lower than approximately 46%. This finding suggests that most Cl species were adsorbed on $CeO_2$. However, $HPW/CeO_2$ exhibited a very high selectivity toward HCl, and its HCl selectivity was close to 100% at 250–450 °C (Figure 1b), which was significantly higher than that of $CeO_2$. This suggests that the selectivity of $CeO_2$ toward HCl was significantly improved after the coating of HPW, and most of the Cl species adsorbed on $HPW/CeO_2$ were able to escape the surface as HCl. Moreover, the stability of $HPW/CeO_2$ for the oxidation of CB was much better than that of $CeO_2$, and the CB conversion efficiency of $HPW/CeO_2$ was almost stable at 70% at 300 °C for 30 h (Figure 2a). Meanwhile, the HCl selectivity of $HPW/CeO_2$ during CB oxidation was basically maintained at 100% for 30 h (Figure S1). These results further demonstrate that the HPW coating could promote the removal of Cl species from $CeO_2$ as HCl, avoiding the occurrence of Cl poisoning. Additionally, the selectivity of $CeO_2$ toward $CO_x$ was also significantly improved after

coating with HPW (Figure 1c), and the $CO_x$ selectivity of HPW/$CeO_2$ during CB oxidation was basically maintained at 100% for 30 h (Figure S1).

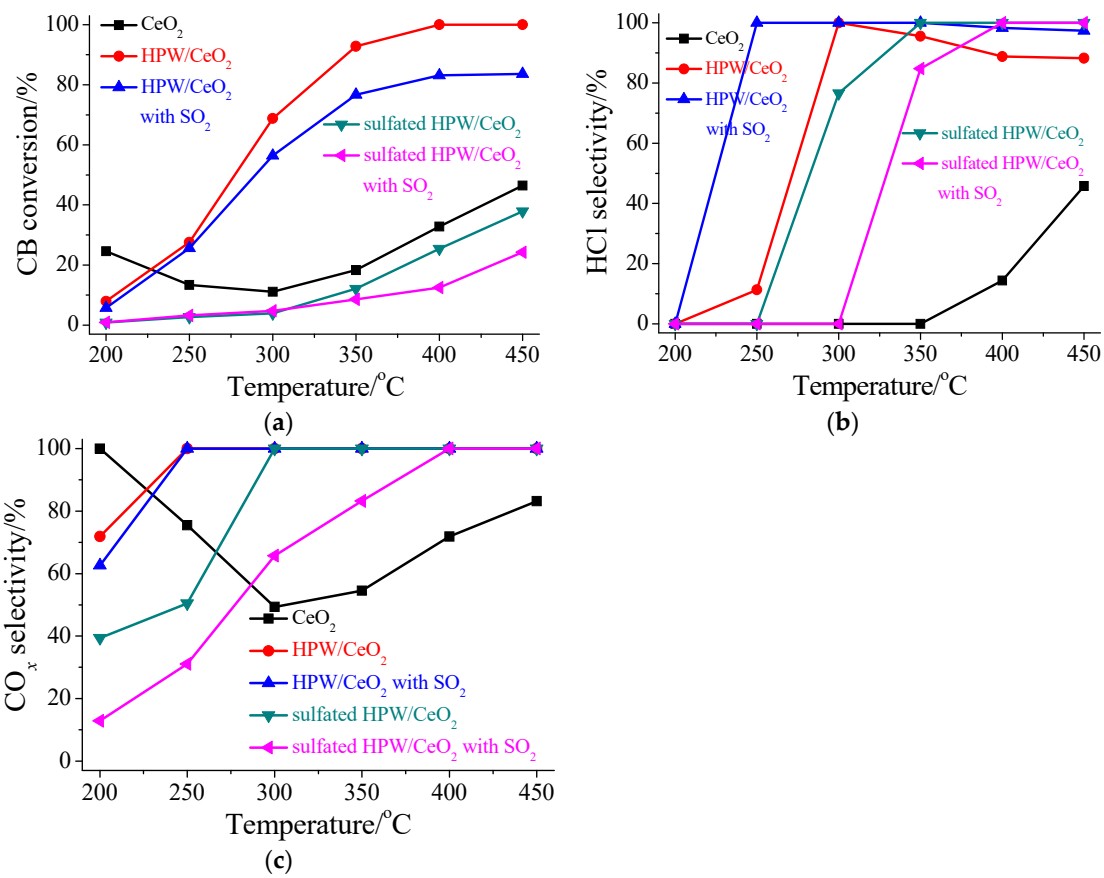

(a)

(b)

(c)

**Figure 1.** (**a**) CB conversion efficiencies, (**b**) HCl selectivities, and (**c**) $CO_2$ selectivities of $CeO_2$, HPW/$CeO_2$, HPW/$CeO_2$ with $SO_2$, sulfated HPW/$CeO_2$, and sulfated HPW/$CeO_2$ with $SO_2$. Operating conditions: [CB] = 100 ppm, [$O_2$] = 5%, [$SO_2$] = 100 ppm (during use), catalyst mass = 200 mg, total flow rate = 200 mL $min^{-1}$, and WHSV = 60,000 $cm^3$ $g^{-1}$ $h^{-1}$.

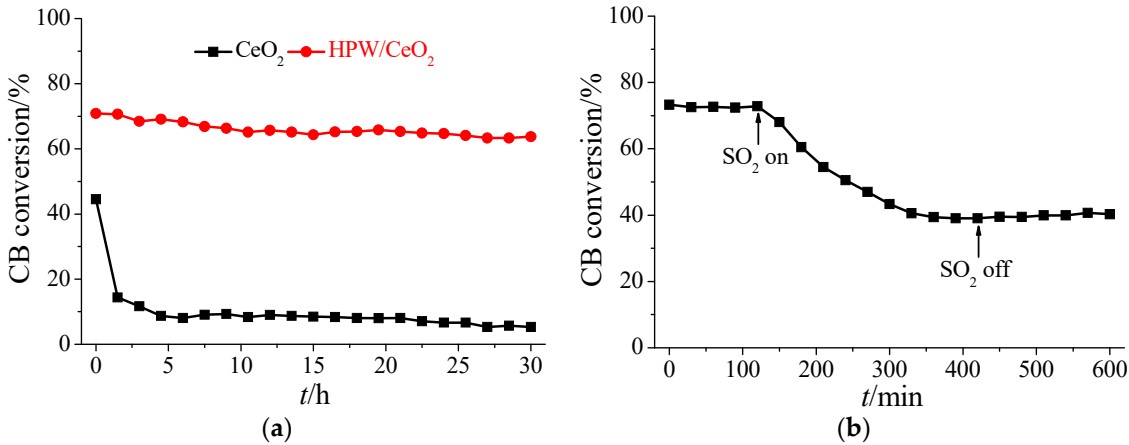

(a)

(b)

**Figure 2.** (**a**) CB conversion efficiencies of $CeO_2$ and HPW/$CeO_2$ at 300 °C for 30 h and (**b**) influence of $SO_2$ on CB oxidation by HPW/$CeO_2$ at 300 °C. Operating conditions: [CB] = 100 ppm, [$O_2$] = 5%, [$SO_2$] = 100 ppm, catalyst mass = 200 mg, total flow rate = 200 mL $min^{-1}$, and WHSV = 60,000 $cm^3$ $g^{-1}$ $h^{-1}$.

### 3.2. Influence of $SO_2$ in the Flue Gas

$SO_2$ is frequently present alongside Cl–VOCs in certain industrial flue gases, such as those produced during steel sintering and waste incineration [4]. Hence, the impact of $SO_2$ on CB oxidation by HPW/CeO$_2$ was investigated. When $SO_2$ was added into the system, the CB conversion efficiency of HPW/CeO$_2$ significantly decreased (as shown in Figure 1a). This discovery indicates that $SO_2$ greatly inhibited CB oxidation by HPW/CeO$_2$. These findings highlight the detrimental effect of $SO_2$ on the activity of HPW/CeO$_2$ in promoting the oxidation of CB.

To further verify the effect of $SO_2$ on the oxidation of CB by HPW/CeO$_2$, an experiment was conducted where $SO_2$ was passed into the reaction mixture at 300 °C. Initially, the CB conversion efficiency of HPW/CeO$_2$ remained stable at approximately 72% for 2 h (Figure 2b). However, upon introducing 100 ppm of $SO_2$, the CB conversion efficiency of HPW/CeO$_2$ started to gradually decrease and reached approximately 40% within 6 h (Figure 2b). This observation clearly indicates that the presence of $SO_2$ significantly restrained the oxidation of CB by HPW/CeO$_2$. Remarkably, even after the supply of $SO_2$ was stopped, the CB conversion efficiency of HPW/CeO$_2$ did not revert to its initial state. This finding suggests that the inhibition of CB oxidation by HPW/CeO$_2$ caused by $SO_2$ was irreversible. The irreversibility of the inhibition might be ascribed to surface sulfation of HPW/CeO$_2$ due to the reaction between $SO_2$ and HPW/CeO$_2$.

To gain a more comprehensive understanding of the impact of surface sulfation on CB oxidation, a comparison was made between the CB oxidation activities of HPW/CeO$_2$ and sulfated HPW/CeO$_2$. The results clearly demonstrate that the CB conversion efficiency of HPW/CeO$_2$ experienced a significant decline after the sulfation process (Figure 1a). Meanwhile, the HCl and CO$_x$ selectivities of sulfated HPW/CeO$_2$ were both significantly lower than those of HPW/CeO$_2$ (Figure 1b,c). These results indicate that the surface sulfation of HPW/CeO$_2$ played a crucial role in impeding the catalytic activity of $SO_2$ on CB oxidation.

### 3.3. Characterization

#### 3.3.1. XRD and BET Surface Area

The XRD patterns of the synthesized CeO$_2$ and HPW/CeO$_2$ (Figure 3a) closely resembled that of cerianite (CeO$_2$, JPCDS 34-00394). This implies that the HPW coating on CeO$_2$ did not introduce significant changes to the cubic fluorite structure of CeO$_2$. Moreover, the XRF analysis revealed that the quantity of HPW in HPW/CeO$_2$ was very small (Table S1). Hence, no other peaks corresponding to HPW or WO$_3$ were detected in the XRD pattern (Figure 3a). Interestingly, even after sulfation, the XRD pattern of HPW/CeO$_2$ remained relatively unaffected (Figure 3a). This observation suggests that the sulfation process did not cause any significant damage to the crystal structure of HPW/CeO$_2$.

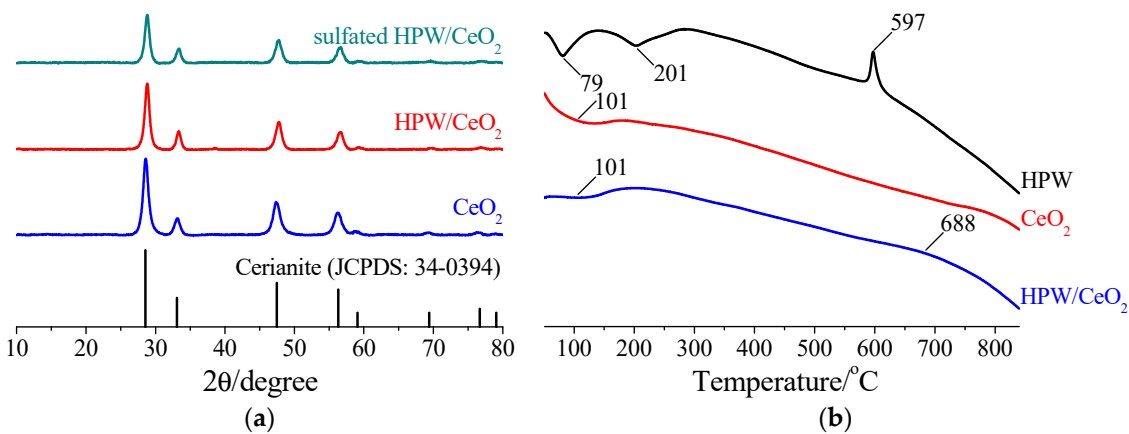

**Figure 3.** (**a**) XRD patterns of CeO$_2$, HPW/CeO$_2$, and sulfated HPW/CeO$_2$ and (**b**) DTA curves of HPW, CeO$_2$, and HPW/CeO$_2$.

The BET surface areas of $CeO_2$, $HPW/CeO_2$, and sulfated $HPW/CeO_2$ were approximately 55.4, 56.4, and 16.2 $m^2$ $g^{-1}$, respectively.

### 3.3.2. DTA

The DTA curve of HPW showed a sharp exothermic peak at 597 °C (Figure 3b), which was attributed to the decomposition of the HPW Keggin structure, leading to the formation of $WO_3$ [22]. Additionally, two distinct endothermic peaks appeared at 79 and 201 °C (Figure 3b), which correspond to the two different stages of dehydration that the HPW underwent [22]. In the case of $CeO_2$, a single endothermic peak appeared at 101 °C (Figure 3b), which was ascribed to the dehydration stage of $CeO_2$ [12]. The DTA curve of $HPW/CeO_2$ showed an endothermic peak at 101 °C and an exothermic peak at 688 °C (Figure 3b), which are related to the dehydration of $CeO_2$ and the decomposition of HPW, respectively. The shift in the exothermic peak from 597 to 688 °C in the $HPW/CeO_2$ system clearly demonstrates that $CeO_2$ significantly enhanced the thermal stability of HPW. This suggests that the incorporation of $CeO_2$ into the system improved the resistance of HPW to decomposition and increased its overall thermal stability.

### 3.3.3. XPS

The Ce 3d binding energies for $HPW/CeO_2$ were observed at 882.6, 884.7, 889.2, 898.6, 901.1, 902.9, 907.7, and 916.9 eV (Figure 4a). The binding energies at 884.7 and 902.9 eV correspond to $Ce^{3+}$ in $Ce_2O_3$, while the binding energies at 882.6, 889.2, 898.6, 901.1, 907.7, and 916.9 eV were attributed to $Ce^{4+}$ in $CeO_2$ [23]. The O 1s binding energies for $HPW/CeO_2$ were measured at 529.4, 530.7, and 532.3 eV (Figure 4b), which are associated with lattice oxygen, adsorbed oxygen, and oxygen in the HPW, respectively [24–26]. The W 4f binding energies for $HPW/CeO_2$ were recorded at 35.5 and 37.6 eV (Figure 4c), which correspond to the W $4f_{7/2}$ and W $4f_{5/2}$ of $W^{6+}$ in the HPW, respectively [27,28]. Therefore, $HPW/CeO_2$ contains both $Ce^{3+}$ and $Ce^{4+}$ species, as well as lattice oxygen and adsorbed oxygen species. Importantly, the Keggin structure of HPW remains intact in $HPW/CeO_2$. Additionally, an analysis of the percentages of W and Ce species on the surface of $HPW/CeO_2$ compared to their content within $HPW/CeO_2$ (Table S1) revealed that the percentage of W species on the surface was significantly higher than that within $HPW/CeO_2$. Meanwhile, EDS mapping demonstrates that W and P elements were clearly observed on $HPW/CeO_2$ (Figure S2). These results suggest that HPW was primarily located on the surface of $CeO_2$, rather than being dispersed evenly throughout the catalyst.

After sulfation, two new binding energies were observed at 886.6 and 905.4 eV in the Ce 2p spectrum of $HPW/CeO_2$ (Figure 4a), which were identified as being associated with $Ce^{3+}$ in $Ce_2(SO_4)_3$ [29]. Additionally, a distinct new O 1s binding energy was detected at 531.9 eV in sulfated $HPW/CeO_2$ (Figure 4b), which was attributed to $O^{2-}$ in $SO_4^{2-}$ [30]. Furthermore, two new S 2p binding energies were observed at 168.5 and 169.8 eV in sulfated $HPW/CeO_2$ (Figure 4d), which were linked to $S^{6+}$ in $SO_4^{2-}$ [30]. These results clearly indicate that the formation of $Ce_2(SO_4)_3$ on $HPW/CeO_2$ occurred through the reaction between $CeO_2$ on $HPW/CeO_2$ and $SO_2$. However, upon analysis of the W 4f spectrum, it was found that sulfated $HPW/CeO_2$ exhibited a similar spectrum to that of $HPW/CeO_2$ (Figure 4c). This indicates that the reaction between HPW on $HPW/CeO_2$ and $SO_2$ was insignificant or negligible.

After conducting CB oxidation for 10 h, an analysis of the Ce 3d, W 4f, and O 1s spectra of $HPW/CeO_2$ revealed that there were no significant changes (Figure 4a–c). Moreover, the spectrum of $HPW/CeO_2$ did not exhibit any detectable peak corresponding to Cl 2p after 10 h of CB oxidation (Figure 4e). This finding strongly suggests that there was only a minimal deposition of Cl species on the surface of $HPW/CeO_2$ during the CB oxidation process. As a result, it can be concluded that $HPW/CeO_2$ exhibited an exceptional resistance to Cl poisoning, which is effectively represented in Figure 2a.

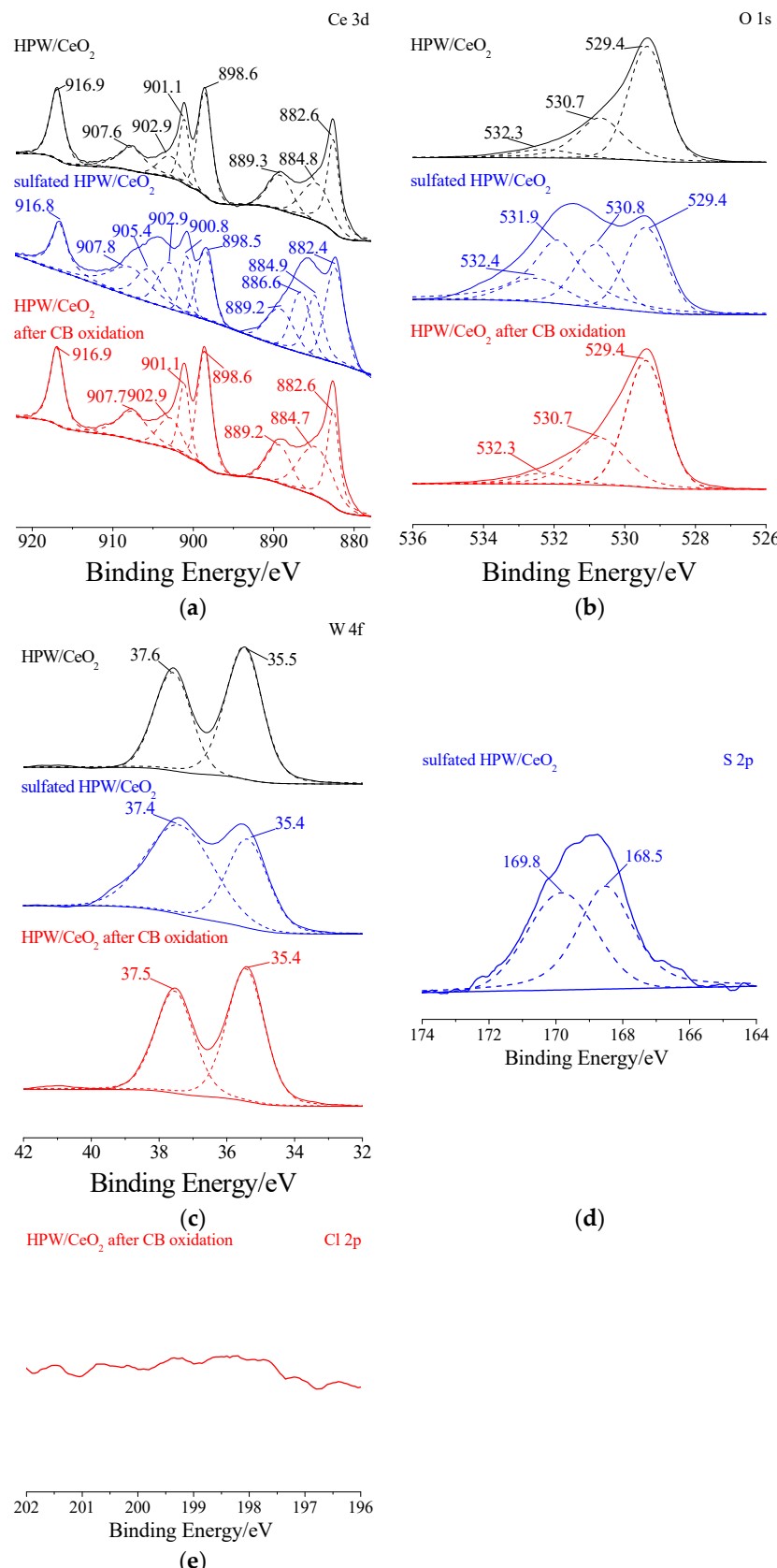

**Figure 4.** XPS spectra of HPW/CeO₂, sulfated HPW/CeO₂, and HPW/CeO₂ after CB oxidation in the spectral regions of (**a**) Ce 3d, (**b**) O 1s, (**c**) W 4f, (**d**) S 2p, and (**e**) Cl 2p.

### 3.3.4. H$_2$–TPR

There were two distinct reduction peaks in the H$_2$–TPR profile of CeO$_2$ (Figure 5). The first peak, which occurred at 523 °C, was found to be associated with the reduction of surface oxygen species [31]. The second peak, occurring at a higher temperature of 777 °C, was attributed to the reduction of oxygen located within the bulk [31]. After coating with HPW, it was observed that the reduction of CeO$_2$ did not significantly change (Figure 5). However, two weaker reduction peaks emerged at 565 and 706 °C (Figure 5). Because Ce$^{3+}$ on the surface is able to reduce W$^{6+}$ on HPW/CeO$_2$, these new peaks were attributed to the indirect reduction of W$^{6+}$ [12]. As the first reduction peak temperatures of CeO$_2$ and HPW/CeO$_2$ remained the same, the coating of HPW did not notably affect the oxidation ability of CeO$_2$. Upon further sulfation, it was observed that the reduction peaks at 523 and 777 °C nearly disappeared (Figure 5). In their places, a new stronger reduction peak appeared at 593 °C in sulfated HPW/CeO$_2$ (Figure 5), which is directly related to the reduction of sulfated species [32]. Interestingly, the significant increase in the temperature required for the first reduction peak suggests that the oxidation ability of HPW/CeO$_2$ was greatly reduced after sulfation.

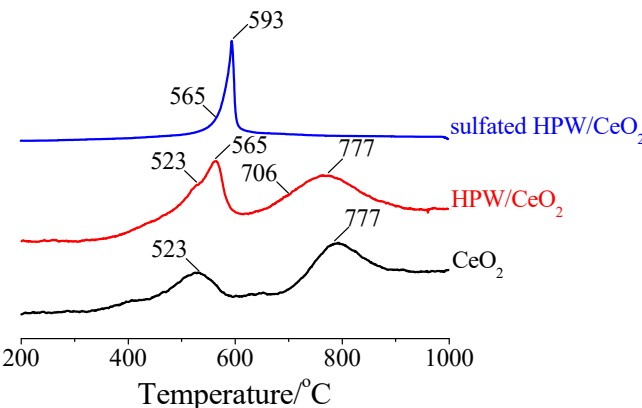

**Figure 5.** H$_2$-TPR profiles of CeO$_2$, HPW/CeO$_2$, and sulfated HPW/CeO$_2$.

### 3.4. Mechanism of CB Oxidation

The possible mechanism of CB oxidation by HPW/CeO$_2$ generally follows the Mars–van Krevelen mechanism (i.e., gaseous CB is firstly adsorbed on the catalyst surface, which is then oxidized by the lattice oxygen species to generate the final product; lastly, gaseous O$_2$ replenishes the lattice oxygen species consumed) and the Eley–Rideal mechanism (i.e., gaseous CB reacts with the adsorbed oxygen species on the surface to generate the final product) [33,34].

CB oxidation by HPW/CeO$_2$ through the Mars–van Krevelen mechanism is approximately described as:

$$C_6H_5Cl_{(g)} \rightarrow C_6H_5Cl_{(ad)} \tag{4}$$

$$C_6H_5Cl_{(ad)} + 28 \equiv Ce^{4+} + 14O^{2-} \rightarrow 6CO_2 + HCl + 2H_2O + 28 \equiv Ce^{3+} \tag{5}$$

$$\equiv Ce^{3+} + O_{2(g)} \rightarrow \equiv Ce^{4+} + 2O^{2-} \tag{6}$$

CB oxidation by HPW/CeO$_2$ through the Eley–Rideal mechanism is approximately described as:

$$O_{2(g)} \rightarrow 2O_{(ad)} \tag{7}$$

$$C_6H_5Cl_{(g)} + 14O_{(ad)} \rightarrow 6CO_2 + HCl + 2H_2O \tag{8}$$

To investigate the role of the Mars–van Krevelen mechanism in CB oxidation by HPW/CeO$_2$, in situ DRIFTS of O$_2$ passing over HPW/CeO$_2$ pre-adsorbed by CB at 100–400 °C was performed. After subjecting HPW/CeO$_2$ to CB at 100 °C for 30 min, four distinctive bands at 1445, 1477, 1581, and 1625 cm$^{-1}$ appeared (Figure 6a). The bands

at 1445 and 1477 cm$^{-1}$ were identified as the stretching vibrations of the C=C bond in CB adsorbed on Brønsted acid sites due to the Ce–OH in $CeO_2$ in $HPW/CeO_2$ [35], and the band at 1477 cm$^{-1}$ was also ascribed to the stretching vibration of the C=C bond in CB adsorbed on Brønsted acid sites due to the W–OH in HPW in $HPW/CeO_2$ [9]. The band at 1581 cm$^{-1}$ corresponds to the stretching vibration of the C=C bond in CB adsorbed on Lewis acid sites, and was created by $Ce^{3+}/Ce^{4+}$ species in $CeO_2$ in $HPW/CeO_2$ [36]. The band at 1625 cm$^{-1}$ represents the out-of-plane bending vibrations of the C–H bond in the aromatic ring of CB adsorbed on Lewis acid sites, associated with $W^{6+}$ species in HPW [33]. These results suggest that CB was adsorbed on $HPW/CeO_2$. As the reaction temperature was increased to 200 °C, the bands related to the adsorbed CB gradually disappeared, while seven new bands at 1302, 1363, 1413, 1530, 1580, 1660, and 1690 cm$^{-1}$ appeared (Figure 6a). The bands at 1302 and 1580 cm$^{-1}$ indicate the formation of phenolate species, represented by the stretching vibrations of the C–O bond in phenolate and the C=C bond in the aromatic ring, respectively [9,37]. These species were generated through the cleavage of the C–Cl bond in CB via a nucleophilic substitution reaction with lattice oxygen species. The bands at 1660 and 1690 cm$^{-1}$ represent the stretching vibrations of the C=O bond in p-benzoquinone and o-benzoquinone species, respectively [38,39]. This suggests that some phenolate species were attacked by lattice oxygen, resulting in the formation of benzoquinone. The band at 1530 cm$^{-1}$ corresponds to the symmetric stretching vibration of the $COO^-$ groups, indicating the presence of maleic anhydride species [40]. This suggests that certain benzoquinone species were attacked by lattice oxygen, leading to the cleavage of the aromatic ring and the formation of maleic anhydride species. The band at 1413 cm$^{-1}$ represents the asymmetric stretching vibration of the $COO^-$ groups [38], suggesting that the maleic anhydride species were further oxidized to form acetate species. The band at 1363 cm$^{-1}$ was assigned to the stretching vibration of the $COO^-$ groups in acetate species [41], indicating that the remaining maleic anhydride species were being further oxidized. When the reaction temperature reached 350 °C, the bands corresponding to phenolate species, benzoquinone species, and maleic anhydride species nearly disappeared. Only one new band at 1605 cm$^{-1}$ was present, which indicates the stretching vibrations of the $COO^-$ groups in acetate species [42]. This suggests that some acetate species underwent further oxidation to form the final products. Overall, these results strongly support the conclusion that CB adsorbed on $HPW/CeO_2$ can be oxidized by lattice oxygen species, ultimately resulting in the formation of the final products. Hence, the Mars–van Krevelen mechanism plays a significant role in the oxidation of CB by $HPW/CeO_2$.

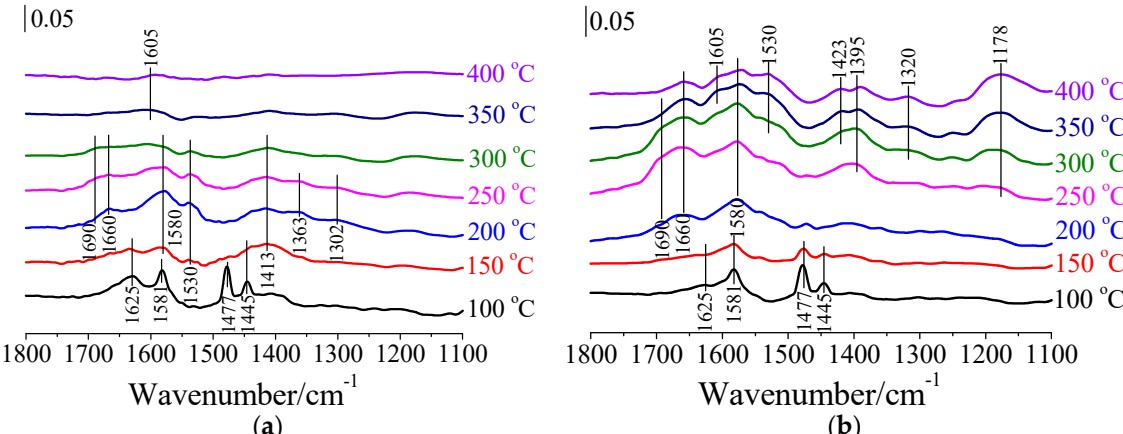

**Figure 6.** (**a**) In situ DRIFTS spectra of $O_2$ passing over $HPW/CeO_2$ pre-adsorbed by CB at 100–400 °C. (**b**) In situ DRIFTS spectra of CB + $O_2$ passing over $HPW/CeO_2$ at 100–400 °C.

According to Reaction (5), the CB oxidation rate of HPW/CeO$_2$ through the Mars–van Krevelen mechanism (i.e., $\delta_{\text{MvK}}$) is approximately described as:

$$\delta_{\text{MvK}} = -\frac{d[\text{C}_6\text{H}_5\text{Cl}_{(g)}]}{dt} = k_1[\text{C}_6\text{H}_5\text{Cl}_{(ad)}][\equiv \text{Ce}^{4+}]^{\alpha}[\text{O}^{2-}]^{\beta} \tag{9}$$

where $k_1$, $[\text{C}_6\text{H}_5\text{Cl}_{(ad)}]$, $[\equiv\text{Ce}^{4+}]$, $[\text{O}^{2-}]$, $\alpha$, and $\beta$ are the kinetic constant of Reaction (5), the amounts of CB adsorbed, Ce$^{4+}$ bonded with O$^{2-}$, and lattice oxygen species on the surface, and reaction orders of Reaction (5) with regard to the amounts of surface Ce$^{4+}$ and O$^{2-}$, respectively.

According to Reaction (8), the CB oxidation rate of HPW/CeO$_2$ through the Eley–Rideal mechanism (i.e., $\delta_{\text{E–R}}$) is approximately described as:

$$\delta_{\text{E–R}} = -\frac{d[\text{C}_6\text{H}_5\text{Cl}_{(g)}]}{dt} = k_2[\text{C}_6\text{H}_5\text{Cl}_{(g)}][\text{O}_{(ad)}]^{\gamma} \tag{10}$$

where $k_2$, $[\text{C}_6\text{H}_5\text{Cl}_{(g)}]$, $[\text{O}_{(ad)}]$, and $\gamma$ are the kinetic constant of Reaction (8), the amounts of gaseous CB in the flue gas and surface adsorbed oxygen species, and the reaction order of Reaction (8) with regard to the amount of surface adsorbed oxygen species, respectively.

Therefore, the CB oxidation rate of HPW/CeO$_2$ is approximately described as:

$$\begin{aligned}\delta &= \delta_{\text{MvK}} + \delta_{\text{E–R}} \\ &= k_1[\text{C}_6\text{H}_5\text{Cl}_{(ad)}][\equiv \text{Ce}^{4+}]^{\alpha}[\text{O}^{2-}]^{\beta} + k_2[\text{C}_6\text{H}_5\text{Cl}_{(g)}][\text{O}_{(ad)}]^{\gamma}\end{aligned} \tag{11}$$

The concentration of CB in the flue gas was generally high (~100 ppm), suggesting that HPW/CeO$_2$ was almost saturated with the adsorption of CB. Hence, the amount of CB adsorbed on HPW/CeO$_2$ can be considered as a constant. Furthermore, because both Ce$^{4+}$ and O$^{2-}$ can be quickly recovered through Reaction (6), the concentrations of Ce$^{4+}$ and O$^{2-}$ on HPW/CeO$_2$ also remained constant. Moreover, the concentration of O$_2$ in the flue gas was approximately 5%, which was about 500 times higher than the concentration of CB. This means that the decrease in the concentration of O$_{(ad)}$ on HPW/CeO$_2$ due to CB oxidation (i.e., Reaction (8)) can be neglected. Thus, the concentration of O$_{(ad)}$ on HPW/CeO$_2$ can be regarded as a constant. As suggested by Equation (10), the CB oxidation rate of HPW/CeO$_2$ should exhibit an excellent linear relationship with the CB concentration. The intercept and slope of this relationship can be used to describe the kinetic constants of CB oxidation through the Mars–van Krevelen mechanism (i.e., $k_{\text{MvK}}$) and the Eley–Rideal mechanism (i.e., $k_{\text{E–R}}$), respectively.

Therefore, Equation (11) can be approximately revised to:

$$\delta = \delta_{\text{MvK}} + \delta_{\text{E–R}} = k_{\text{MvK}} + k_{\text{E–R}}[\text{C}_6\text{H}_5\text{Cl}_{(g)}] \tag{12}$$

To determine the kinetic constants of CB oxidation using the Mars–van Krevelen mechanism and the Eley–Rideal mechanism, a kinetics experiment of CB oxidation by HPW/CeO$_2$ at 250–450 °C with a CB conversion efficiency lower than 15% was performed, and the dependence of the CB conversion rate of HPW/CeO$_2$ on the CB concentration is shown in Figure 7. Figure 7 shows that the CB conversion rate of HPW/CeO$_2$ increases significantly as the CB concentration increases. The results reveal a linear relationship between the CB oxidation rate and the CB concentration, indicating a direct dependence. This finding aligns with the assumption stated in Equation (12), validating its accuracy. To gain further insights from the data, a linear regression analysis was conducted on Figure 7, utilizing Equation (12) as the foundational equation. This analysis involved determining the slope, intercept, and regression coefficient of the linear regression equation, which are all listed in Table 1.

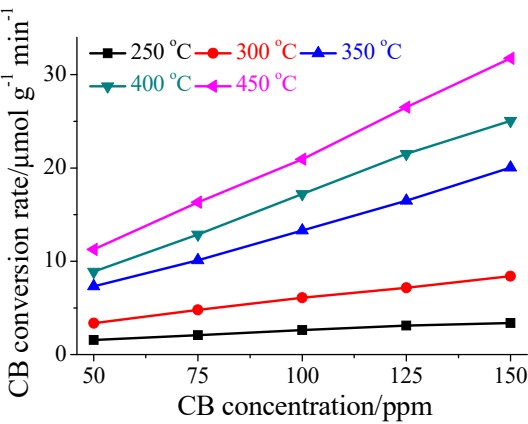

**Figure 7.** Dependence of the CB conversion rate of HPW/CeO$_2$ on the CB concentration. Operating conditions: [CB] = 50–150 ppm, [O$_2$] = 5%, catalyst mass = 3.8–30 mg, total flow rate = 200 mL min$^{-1}$, and WHSV = 400,000–3,160,000 cm$^3$ g$^{-1}$ h$^{-1}$.

**Table 1.** Reaction kinetic constants of CB oxidation by HPW/CeO$_2$.

| | Temperature/°C | /µmol g$^{-1}$ min$^{-1}$ | | |
| --- | --- | --- | --- | --- |
| | | $k_{E-R}$ | $k_{MvK}$ | R$^2$ |
| HPW/CeO$_2$ | 250 | 0.00939 | 1.71 | 0.999 |
| | 300 | 0.0347 | 2.81 | 0.999 |
| | 350 | 0.0961 | 3.56 | 0.998 |
| | 400 | 0.157 | 4.91 | 0.996 |
| | 450 | 0.213 | 4.16 | 0.999 |

According to the data presented in Table 1, the values of the intercept ($k_{MvK}$) were determined to be approximately 1.71, 2.81, 3.56, 4.91, and 4.16 µmol g$^{-1}$ min$^{-1}$ at temperatures of 250, 300, 350, 400, and 450 °C, respectively. Table 1 also indicates that the values of the slope ($k_{E-R}$) were around 0.00939, 0.0347, 0.0961, 0.157, and 0.213 µmol g$^{-1}$ min$^{-1}$ at the same temperatures. These observations provide evidence that the oxidation of CB by HPW/CeO$_2$ was influenced by both the Mars–van Krevelen mechanism and the Eley–Rideal mechanism. Furthermore, it was discovered that the rate of CB oxidation by HPW/CeO$_2$ through the Eley–Rideal mechanism was directly proportional to the CB concentration, which was supported by Equation (10). Therefore, when the CB concentration was around 182, 81, 37, 31, and 20 ppm at temperatures of 250, 300, 350, 400, and 450 °C, respectively, the CB oxidation rate through the Eley–Rideal mechanism was equivalent to that through the Mars–van Krevelen mechanism. Therefore, these two mechanisms contributed equally to CB oxidation by HPW/CeO$_2$ at these specific CB concentrations. However, when the CB concentration fell below the aforementioned values at each temperature, the CB oxidation rate through the Mars–van Krevelen mechanism surpassed that of the Eley–Rideal mechanism. This implies that the Mars–van Krevelen mechanism plays a more dominant role in CB oxidation by HPW/CeO$_2$ at lower CB concentrations. On the other hand, when the CB concentration exceeded the stated values, the CB oxidation rate through the Eley–Rideal mechanism exceeded that of the Mars–van Krevelen mechanism. This suggests that the Eley–Rideal mechanism becomes more significant in CB oxidation by HPW/CeO$_2$ at higher CB concentrations. Considering the fact that the CB concentration in the flue gas typically reached approximately 100 ppm, the CB oxidation rate of HPW/CeO$_2$ through the Mars–van Krevelen mechanism was greater than that of the Eley–Rideal mechanism at 250 °C. However, at temperatures ranging from 300 to 450 °C, the CB oxidation rate through the Eley–Rideal mechanism was greater than that of the Mars–van Krevelen mechanism. Therefore, the oxidation of CB by HPW/CeO$_2$ was influenced by both the temperature and the CB concentration. The Mars–van Krevelen mechanism appears to be

more important at lower temperatures and lower CB concentrations, while the Eley–Rideal mechanism is more dominant at higher temperatures and higher CB concentrations.

To gain a deeper understanding of the reaction pathway of CB oxidation by HPW/CeO$_2$ through the Eley–Rideal mechanism, in situ DRIFTS of CB + O$_2$ passing over HPW/CeO$_2$ was conducted at 100–400 °C. In addition to the known bands corresponding to adsorbed CB at 1445, 1477, 1581, and 1625 cm$^{-1}$, phenolate species at 1580 cm$^{-1}$, benzoquinone species at 1660 and 1690 cm$^{-1}$, maleic anhydride species at 1530 cm$^{-1}$, acetate species at 1605 cm$^{-1}$, and four new bands at 1178, 1320, 1395, and 1423 cm$^{-1}$ appeared in the HPW/CeO$_2$ spectrum. The bands at 1178 and 1320 cm$^{-1}$ are associated with the symmetric stretching vibration of the COO$^-$ groups originating from maleic anhydride species [43,44]. The band at 1395 cm$^{-1}$ was attributed to the stretching vibration of the –CH$_2$– bond in acetate species [41]. Lastly, the band at 1423 cm$^{-1}$ was assigned to the stretching vibration of the COO$^-$ groups from acetate species [45]. These results suggest that the intermediates formed during the CB oxidation process on HPW/CeO$_2$ through the Eley–Rideal mechanism mainly consists of phenolate species, benzoquinone species, maleic anhydride species, and acetate species. Therefore, the reaction pathway of CB oxidation by HPW/CeO$_2$ via the Eley–Rideal mechanism may either be the same as or simplified compared to that observed in the Mars–van Krevelen mechanism.

According to the analysis results of in situ DRIFTS and reaction kinetics, it was discovered that the oxidation of CB by HPW/CeO$_2$ mainly follows two main mechanisms: the Mars–van Krevelen mechanism and the Eley–Rideal mechanism. Therefore, a reasonable reaction pathway for CB oxidation by HPW/CeO$_2$ is summarized in Figure 8. (1) Initially, a fraction of gaseous CB molecules is adsorbed onto the Brønsted acid sites of HPW and CeO$_2$, as well as the Lewis acid sites of CeO$_2$ on HPW/CeO$_2$. (2) The C–Cl bonds in some of absorbed CB molecules, as well as most of gaseous CB molecules, undergo a cleavage reaction through nucleophilic substitution with the lattice oxygen species. This reaction results in the formation of phenolate species. (3) The phenolate species are subsequently attacked by the lattice oxygen species through an electrophilic substitution reaction, leading to the formation of benzoquinone species. (4) The aromatic ring present in the benzoquinone species is cleaved by the attack of the lattice oxygen species, resulting in the formation of maleic anhydride species. (5) The maleic anhydride species undergoes further deep oxidation, starting with the transformation into acetate species and ultimately oxidizing into CO$_2$, CO, and H$_2$O. Additionally, any Cl species present on HPW/CeO$_2$ are rapidly eliminated through a potential reaction: the dissociatively adsorbed Cl reacts with surface hydroxyl groups from HPW on HPW/CeO$_2$, forming HCl.

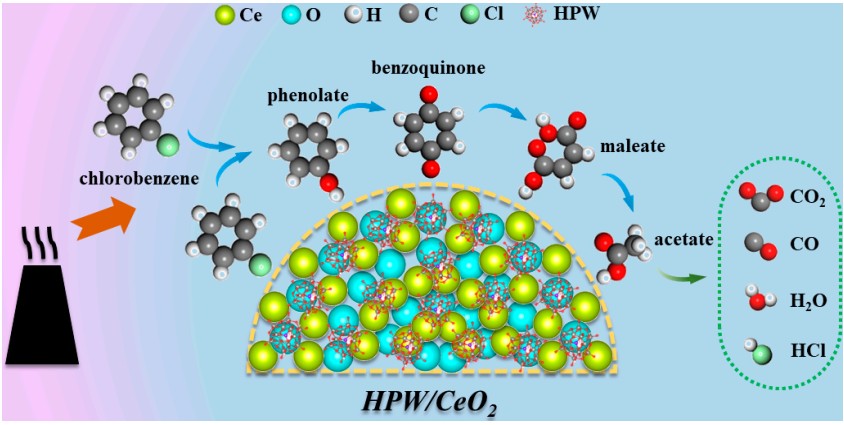

**Figure 8.** A reasonable reaction pathway of CB oxidation by HPW/CeO$_2$.

### 3.5. Synergistic Effect of HPW and CeO₂ on CB Oxidation

Equation (11) indicates that the rate of CB oxidation mainly depends on the oxidation ability of the catalyst and the amounts of surface-adsorbed CB, $Ce^{4+}$ bonded with $O^{2-}$, lattice oxygen species, gaseous CB, and surface-adsorbed oxygen species.

A $H_2$–TPR analysis (Figure 5) revealed that $CeO_2$ has an excellent oxidation ability. Meanwhile, $CeO_2$ also generally contains abundant $Ce^{4+}$ species bonded with $O^{2-}$, lattice oxygen species, and adsorbed oxygen species on the surface. Furthermore, a CB–TPD analysis (Figure 9a) indicated that the amount of CB adsorbed on $CeO_2$ was high (~33.9 μmol g⁻¹). However, $CeO_2$ is susceptible to Cl poisoning during CB oxidation (Figure S3), resulting in remarkable decreases in the oxidation ability; in the amounts of $Ce^{4+}$ bonded with $O^{2-}$, lattice oxygen species, and adsorbed oxygen species on the surface; and in the amount of adsorbed CB. Therefore, $CeO_2$ exhibited poor performance in CB oxidation (Figure 1a).

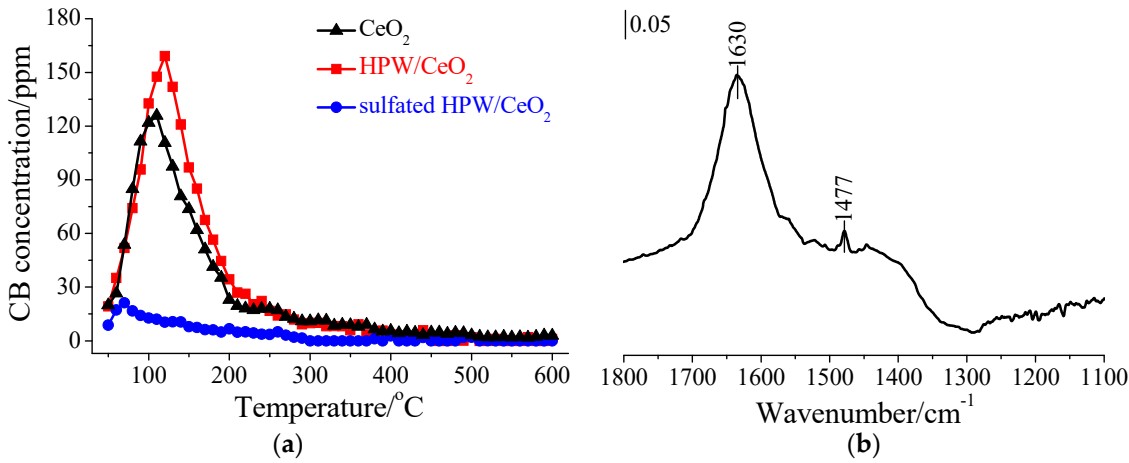

**Figure 9.** (**a**) CB–TPD profiles of $CeO_2$, HPW/$CeO_2$, and sulfated HPW/$CeO_2$ and (**b**) in situ DRIFTS spectrum of CB passing over sulfated HPW/$CeO_2$ at 100 °C for 30 min.

A CB–TPD analysis showed that the amount of CB adsorbed on HPW/$CeO_2$ was around 37.9 μmol g⁻¹, which was slightly larger than that adsorbed on $CeO_2$ (Figure 9a). This implies that the amount of CB adsorbed on $CeO_2$ slightly increased after coating with HPW, and the amount of CB adsorbed on HPW/$CeO_2$ was mostly high. Meanwhile, a $H_2$–TPR analysis indicated that HPW/$CeO_2$ showed excellent oxidation ability, which was similar to that of $CeO_2$ (Figure 5). As suggested by Equation (11), HPW/$CeO_2$ exhibits an excellent performance in CB oxidation (Figure 1a), despite the slight decreases in the amounts of $Ce^{4+}$ bonded with $O^{2-}$, lattice oxygen species, and adsorbed oxygen species on $CeO_2$ due to the coverage of HPW. However, HPW exhibited a poor performance in CB oxidation (Figure S4). As a result, HPW plays an important role in providing abundant Brønsted acid sites for the removal of Cl species from $CeO_2$ as HCl, resulting in the high HCl selectivity of HPW/$CeO_2$ (Figure 1b). This not only prevents Cl poisoning of $CeO_2$ during CB oxidation but also results in a superior CB oxidation performance. Overall, the combination of HPW and $CeO_2$ proved to be an effective catalyst for CB oxidation.

### 3.6. Inhibition Mechanism of SO₂ on CB Oxidation

The CB oxidation rate of HPW/$CeO_2$ is closely related to its oxidation ability; the amounts of adsorbed CB, $Ce^{4+}$ bonded with $O^{2-}$, lattice oxygen species, and adsorbed oxygen species on its surface; and the concentration of gaseous CB (as shown in Equation (11)). However, the CB concentration in the flue gas is minimally affected by $SO_2$. Therefore, the inhibition of $SO_2$ on CB oxidation by HPW/$CeO_2$ may be related to the following several factors: (1) The decrease in oxidation ability, which mainly results from surface sulfation due to the reaction between $SO_2$ and HPW/$CeO_2$. (2) The inhibition of CB adsorption,

which is mainly related to the decrease in the amount of adsorption sites due to surface sulfation or competitive adsorption between $SO_2$ and CB. (3) The decrease in the amounts of $Ce^{4+}$ bonded with $O^{2-}$, lattice oxygen species, and adsorbed oxygen species, which is mainly attributed to surface sulfation.

To investigate the effect of $SO_2$ on the oxidation ability of $HPW/CeO_2$, the $H_2$–TPR profiles of $HPW/CeO_2$ and sulfated $HPW/CeO_2$ were compared. Figure 5 shows that the oxidation ability of sulfated $HPW/CeO_2$ was remarkably lower than that of $HPW/CeO_2$. This finding implies that the oxidation ability of $HPW/CeO_2$ was significantly reduced by $SO_2$ due to surface sulfation.

To investigate the influence of $SO_2$ on the adsorption sites of $HPW/CeO_2$ for CB adsorption, the CB–TPD profiles of $HPW/CeO_2$ and sulfated $HPW/CeO_2$ were compared. Figure 9a shows that the amount of CB adsorbed on sulfated $HPW/CeO_2$ was around 5.3 μmol g$^{-1}$, which only accounts for approximately 14% of the amount of CB adsorbed on $HPW/CeO_2$ (~37.9 μmol g$^{-1}$). This finding suggests that the amount of adsorption sites on $HPW/CeO_2$ for CB adsorption was significantly reduced by $SO_2$ due to surface sulfation. To further ascertain the effect of surface sulfation on the adsorption sites on $HPW/CeO_2$ for CB adsorption, in situ DRIFTS of CB adsorption onto sulfated $HPW/CeO_2$ was performed. Figure 9b shows that only two distinct bands at 1477 and 1630 cm$^{-1}$ appear in the sulfated $HPW/CeO_2$ spectrum after CB adsorption. The band at 1477 cm$^{-1}$ was attributed to the stretching vibration of the C=C bond in CB adsorbed on Brønsted acid sites due to Ce–OH in $CeO_2$ and W–OH in HPW [9], while that at 1630 cm$^{-1}$ was ascribed to the stretching vibrations of the C=C bond in CB adsorbed on the sulfated species [46]. However, compared with the in situ DRIFTS spectrum of CB adsorption onto $HPW/CeO_2$ (Figure 6a), the bands at 1445, 1581, and 1625 cm$^{-1}$ corresponding to the Brønsted acid sites due to Ce–OH in $CeO_2$, Lewis acid sites due to $Ce^{3+}/Ce^{4+}$ species in $CeO_2$, and Lewis acid sites due to $W^{6+}$ species in HPW for CB adsorption all disappeared. These results further demonstrate that $SO_2$ can significantly decrease the adsorption sites on $HPW/CeO_2$ for CB adsorption due to surface sulfation. Additionally, to investigate whether $SO_2$ competes with CB for adsorption sites, the effect of $SO_2$ on the activity of sulfated $HPW/CeO_2$ for CB oxidation was analyzed. Figure 1a shows that the CB conversion efficiency of sulfated $HPW/CeO_2$ slightly decreased in the presence of $SO_2$. This suggests that $SO_2$ can compete with CB for the adsorption sites on $HPW/CeO_2$, thus inhibiting the adsorption of CB. Hence, $SO_2$ not only sulfates the surface of $HPW/CeO_2$, thereby reducing adsorption sites for CB adsorption, but also competes with CB for adsorption sites. Therefore, the adsorption of CB onto $HPW/CeO_2$ was remarkably restrained by $SO_2$.

To investigate the effect of $SO_2$ on the amounts of $Ce^{4+}$ bonded with $O^{2-}$, lattice oxygen species, and adsorbed oxygen species on $HPW/CeO_2$, the percentages of $Ce^{4+}$ bonded with $O^{2-}$, lattice oxygen species, and adsorbed oxygen species on $HPW/CeO_2$ and sulfated $HPW/CeO_2$ were compared. The percentages of $Ce^{4+}$ bonded with $O^{2-}$, lattice oxygen species, and adsorbed oxygen species on $HPW/CeO_2$ and sulfated $HPW/CeO_2$, as determined via XPS analyses (Figure 4), are listed in Table 2. Table 2 shows that the percentages of $Ce^{4+}$ bonded with $O^{2-}$, lattice oxygen species, and adsorbed oxygen species on sulfated $HPW/CeO_2$ were all significantly smaller than those of $HPW/CeO_2$. This suggests that the amounts of $Ce^{4+}$ bonded with $O^{2-}$, lattice oxygen species, and adsorbed oxygen species on $HPW/CeO_2$ significantly decreased after surface sulfation. Therefore, $SO_2$ can significantly decrease the amounts of $Ce^{4+}$ bonded with $O^{2-}$, lattice oxygen species, and adsorbed oxygen species on $HPW/CeO_2$ due to surface sulfation.

**Table 2.** Percentages of Ce and O species on $HPW/CeO_2$ and sulfated $HPW/CeO_2$/%.

| | **Ce Species** | | | **O Species** | | | |
|---|---|---|---|---|---|---|---|
| | $Ce^{3+}$–O | $Ce^{4+}$–O | $Ce^{4+}$–$SO_4{}^{2-}$ | $O_{lat}$ | $O_{ad}$ | HPW | $SO_4{}^{2-}$ |
| $HPW/CeO_2$ | 4.0 | 16.8 | - | 46.1 | 24.2 | 0.13 | - |
| sulfated $HPW/CeO_2$ | 4.4 | 7.9 | 3.1 | 21.0 | 17.1 | 0.22 | 7.0 |

As $SO_2$ reacts with $HPW/CeO_2$, its surface became sulfated, thus reducing its oxidation ability, the amount of adsorption sites on its surface for CB adsorption, and the amounts of $Ce^{4+}$ bonded with $O^{2-}$, lattice oxygen species, and adsorbed oxygen species on its surface. Meanwhile, $SO_2$ easily competes with CB for the adsorption sites on $HPW/CeO_2$. Hence, CB oxidation by $HPW/CeO_2$ was remarkably restrained by $SO_2$ (as suggested by Equation (11)).

**4. Conclusions**

In this work, HPW was coated on $CeO_2$ to further improve its activity and stability for CB oxidation. Although HPW had a poor activity in CB oxidation, the HPW coating not only facilitated the adsorption of CB onto $CeO_2$, but also provided abundant Brønsted acid sites to $CeO_2$ for the removal of Cl species as HCl, effectively preventing Cl poisoning. Therefore, HPW and $CeO_2$ exhibited a synergistic effect in CB oxidation, and $HPW/CeO_2$ showed superior activity and stability in CB oxidation. The $CeO_2$ in $HPW/CeO_2$ can react with $SO_2$, and thus its surface was easily sulfated when $SO_2$ was present, reducing its oxidation ability; the amount of adsorption sites on its surface for CB adsorption; and the amounts of $Ce^{4+}$ bonded with $O^{2-}$, lattice oxygen species, and adsorbed oxygen species on its surface. Meanwhile, $SO_2$ competed with CB for the adsorption sites on $HPW/CeO_2$. Therefore, the oxidation of CB by $HPW/CeO_2$ was remarkably restrained in the presence of $SO_2$. Importantly, the present work could provide guidance for the design of high-performance Ce-based catalysts for Cl–VOC removal. Meanwhile, it also provides a theoretical basis for improving the $SO_2$ resistance of Ce-based catalysts for Cl–VOC removal.

**Supplementary Materials:** The following supporting information can be downloaded at: https://www.mdpi.com/article/10.3390/su16062245/s1, Figures S1–S6: Selectivities of HCl and $CO_x$ during CB oxidation over $HPW/CeO_2$ at 300 °C for 30 h, STEM image and corresponding EDS mapping of $HPW/CeO_2$, XPS spectra of $CeO_2$ after CB oxidation in the spectral regions of Ce 3d, O 1s, and Cl 2p, CB conversion efficiency of HPW, in situ DRIFTS spectra of passing CB over $CeO_2$ and HPW at 100 °C for 30 min, and CB conversion efficiency of $HPW/CeO_2$ under normal flue gas condition at a low WHSV; Tables S1 and S2: Percentages of Ce and W species on/in $HPW/CeO_2$, and comparison of the performance of $HPW/CeO_2$ for CB oxidation with other reported catalysts. References [47–54] are cited in Supplementary Materials.

**Author Contributions:** Investigation, K.J., L.D., Q.S., W.W. and X.W.; Writing—original draft, J.M.; Writing—review & editing, S.Y. All authors have read and agreed to the published version of the manuscript.

**Funding:** This study was supported by the National Natural Science Foundation of China (Grant 21906070).

**Institutional Review Board Statement:** Not applicable.

**Informed Consent Statement:** Not applicable.

**Data Availability Statement:** Data are contained within the article.

**Conflicts of Interest:** The authors declare no conflict of interest.

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
