# Peer review of "Chlorobenzene Oxidation over Phosphotungstic-Acid-Coated Cerium Oxide: Synergistic Effect of Phosphotungstic and Cerium Oxide and Inhibition Mechanism of Sulfur Dioxide"

_sustainability, doi:10.3390/su16062245_

Round 1

Reviewer 1 Report

Comments and Suggestions for Authors

The authors utilized a serious of experimental techniques such as XRD, XPS, XRF, DRIFTS to investigate the chlorobenzene oxidation mechanism on the surfaces of CeO2 or phosphotungstic acid coated CeO2 as well as the inhibition effects of the produced HCl and of SO2 gas in high temperatures. The manuscript is well structured and presented, therefore, is suitable to be published, after addressing the following minor issues.

1). It would be helpful to have a short paragraph to give an overall structure of the paper at the end of introduction section.

2). The sentence ``...the inhibition mechanism of SO2 on CB oxidation by HPW/CeO2 was also investigated.'' in the last paragraph on page 2 and the following sentences are not coherent. It would be better to revise this part.

3). The CB conversion of HPW/CeO2 with SO2 is around 55% at 300 degree shown in Fig 1a, whereas it becomes 40% after 400 min in Fig. 2b. I assume the SO2 fluxing time is shorter than 400 min and around 200 min for Fig. 1, is it correct? What is the reason to use this elapsed time? Besides, it is necessary to indicate the time in the caption of Fig. 1. In addition, the caption for Fig. 2a is not complete, which should be ``...efficiencies of CeO2 and HPW/CeO2...''. 

4). I suspect that either the number in ``...observed at 95 and...'' on page 6 or the number ``79'' written in Fig. 3b is wrong. 

5). The numbers 37.9 and 5.3 \mumol g^-1 cited on pages 13 and 14 should be specified at a given temperature. I guess they are not corresponding to 300 degree from Fig. 9a. Why don't the authors refer to this temperature when the CB conversion is high enough for HPW/CeO2 shown in Fig. 1?

6). Some abbreviations need the full names, such as ``BET'' and on page 3 and ``DTA'' on page 6. 

Reviewer 2 Report

Comments and Suggestions for Authors

File Attached 

Comments on the Quality of English Language

The manuscript would benefit from improvements in English language proficiency to enhance overall readability and clarity.

Reviewer 3 Report

Comments and Suggestions for Authors

The abstract presented by the authors clearly outlines the issue of Ce-based catalysts generally exhibiting poor activity and stability and proposes the advantages of HPW and the synergistic effect with HPW/CeO2. However, the role of SO2 needs to be clearly described, which needs improvement. The theoretical idea and proposal are clear and well-defined, yet the abstract needs tangible results or a quantitative comparison between methods, and the conclusion needs to be included. I encourage the authors to address these comments to enhance their abstract.

The keywords are redundant with the title, and the use of abbreviations complicates the terminology. Please review your document and the journal's template, as the citation format should be [1] in brackets.

The authors have fascinatingly presented the physicochemical aspects of the oxidation model, the advantages of the synergistic effect of HPW/CeO2, and clarified the mechanism of SO2 involvement, which was unclear in the abstract. Up to this point, the article and introduction are precise. However, the application aspect sought by the journal needs to be reflected. It's revealed that the research question and objectives are directed towards the optimization and evaluation of HPW/CeO2. Therefore, I encourage the authors to dedicate two or three ideas that can contribute to the proof of concept of their hypothesis in environmental aspects and their research contribution.

The results and discussion presented by the authors are robust and appropriate.

The conclusion of the work is pertinent to the results. However, as it is an innovative work, the authors should propose a perspective for the research line that allows researchers to follow the trajectory to enrich the literature.

The authors should review the template's formatting, including citations and sections.

Overall, the work is fascinating, novel, robust, and meets the journal's scope. The authors have correctly posed their research question and proven their hypothesis. I find some details necessary, such as the applicative aspects, to prevent the manuscript from being entirely basic science and thus have an application of their work. I encourage the authors to improve in this regard. Therefore, I propose a subsequent review. For now, it should be accepted with minor corrections.

Reviewer 4 Report

Comments and Suggestions for Authors

This paper investigates catalytic oxidation of chlorobenzene over phosphotungstic acid coated CeO2. The coating of HPW not only promoted the adsorption of CB onto CeO2, but also provided Brønsted acid sites to CeO2 for the Cl species removal as HCl, and thus avoiding Cl poisoning. From the experimental and characterization results, the synergistic effect of HPW and CeO2 on HPW/CeO2 was observed, resulting in superior CB oxidation activity and stability. Additionally, the inhibition mechanism of SO2 on CB oxidation by HPW/CeO2 was deeply investigated to help improve the sulfur resistance of the catalyst in the future. The work is scientifically sound, but some issues should be carefully considered, and specific comments are listed as follows:

1.      The description of background in the introduction was too cumbersome. It is suggested to simplify it. And the author should give a general description of the work he has done in the introduction, while the author only introduced the part of his own research mechanism.

2.      The author measured the specific surface area of the catalysts, but did not introduce how to measure it. The specific surface area of the catalyst to explain what is not described in detail. It is recommended to refer and cite the following relevant literature, such as 10.1016/j.mcat.2022.112689.

3.      Written in the DTA characterization description: “Additionally, two distinct endothermic peaks were observed at 95 and 201 °C (Fig. 3b), which corresponded to the two different stages of dehydration that the HPW underwent.”, which is inconsistent with the content marked in Fig. 3b.

4.      The conclusion that HPW was primarily located on the surface of CeO2 was obtained by XPS. It is recommended to supplement the X-ray (EDX) mapping and STEM-EDS characterization to illustrate the conclusion. It is recommended to refer the following relevant literature, such as 10.1016 / j.mcat.2022.112689

5.      The author did not describe the amount of HPW load, and whether the amount of load affects the reaction performance. In addition, whether the Keggin structure of the loaded HPW can be maintained, it is recommended to add characterization to prove and to refer the following relevant literature, such as 10.1016 / j.mcat.2018.03.005, 10.1016 / j.mcat.2020.111334.

6.      The quality of the image should be further improved, I suggest the author carefully modified, such as XPS mapping can refer to the following relevant literature, such as 10.1016 / j.apcatb.2020.119803.

7.      It is suggested that the analysis of XPS spectra of CeO2 after conducting the CB oxidation should be added to make the conclusion more convincing.

Comments on the Quality of English Language

Minor editing of English language required

Round 2

Reviewer 1 Report

Comments and Suggestions for Authors

The revised manuscript has addressed all my concerns and can be published as it is. 

Author Response

Thanks.

Reviewer 2 Report

Comments and Suggestions for Authors

Completed

Author Response

Thanks.

Reviewer 3 Report

Comments and Suggestions for Authors

The authors have made substantial improvements to their abstract. The introduction is now robust and well-founded. Results are presented and well-structured, and the discussion aligns with the level of the proposal, effectively reflecting the study's objectives and findings. The authors have provided an excellent perspective on their research, demonstrating a thorough understanding and meaningful contribution to their field of study. Given the quality of revisions and the strength of the manuscript, I have no further comments and thus recommend its acceptance.

Author Response

Thanks.

Reviewer 4 Report

Comments and Suggestions for Authors

This paper investigates catalytic oxidation of chlorobenzene over phosphotungstic acid coated CeO2. The coating of HPW not only promoted the adsorption of CB onto CeO2, but also provided Brønsted acid sites to CeO2 for the Cl species removal as HCl, and thus avoiding Cl poisoning. From the experimental and characterization results, the synergistic effect of HPW and CeO2 on HPW/CeO2 was observed, resulting in superior CB oxidation activity and stability. Additionally, the inhibition mechanism of SO2 on CB oxidation by HPW/CeO2 was deeply investigated to help improve the sulfur resistance of the catalyst in the future. The work is scientifically sound, but some issues should be carefully considered, and specific comments are listed as follows:

1.     The description of background in the introduction was too cumbersome. It is suggested to simplify it. And the author should give a general description of the work he has done in the introduction, while the author only introduced the part of his own research mechanism.

2.     The author measured the specific surface area of the catalysts, but did not introduce how to measure it. The specific surface area of the catalyst to explain what is not described in detail. It is recommended to refer and cite the following relevant literature, such as 10.1016/j.mcat.2022.112689.

3.     Written in the DTA characterization description: “Additionally, two distinct endothermic peaks were observed at 95 and 201 °C (Fig. 3b), which corresponded to the two different stages of dehydration that the HPW underwent.”, which is inconsistent with the content marked in Fig. 3b.

4.     The conclusion that HPW was primarily located on the surface of CeO2 was obtained by XPS. It is recommended to supplement the X-ray (EDX) mapping and STEM-EDS characterization to illustrate the conclusion. It is recommended to refer the following relevant literature, such as 10.1016 / j.mcat.2022.112689

5.     The author did not describe the amount of HPW load, and whether the amount of load affects the reaction performance. In addition, whether the Keggin structure of the loaded HPW can be maintained, it is recommended to add characterization to prove and to refer the following relevant literature, such as 10.1016 / j.mcat.2018.03.005, 10.1016 / j.mcat.2020.111334.

6.     The quality of the image should be further improved, I suggest the author carefully modified, such as XPS mapping can refer to the following relevant literature, such as 10.1016 / j.apcatb.2020.119803.

7.     It is suggested that the analysis of XPS spectra of CeO2 after conducting the CB oxidation should be added to make the conclusion more convincing.

Comments on the Quality of English Language

Minor editing of English language required
